# Reduced Quality of Life and Sexual Satisfaction in Isolated Hypogonadotropic Hypogonadism

**DOI:** 10.3390/jcm10122622

**Published:** 2021-06-14

**Authors:** Małgorzata Kałużna, Pola Kompf, Michał Rabijewski, Jerzy Moczko, Jarosław Kałużny, Katarzyna Ziemnicka, Marek Ruchała

**Affiliations:** 1Department of Endocrinology, Metabolism and Internal Medicine, Poznan University of Medical Sciences, 60-355 Poznan, Poland; pola.kompf@gmail.com (P.K.); kaziem@ump.edu.pl (K.Z.); mruchala@ump.edu.pl (M.R.); 2Centre of Postgraduate Medical Education, Department of Reproductive Health, 01-004 Warsaw, Poland; mirab@cmkp.edu.pl; 3Department of Computer Science and Statistics, Poznan University of Medical Sciences, 60-806 Poznan, Poland; jmoczko@ump.edu.pl; 4Department of Otolaryngology, Head and Neck Surgery, Poznan University of Medical Sciences, 60-355 Poznan, Poland; jarek.kaluzny@gmail.com

**Keywords:** isolated hypogonadotropic hypogonadism (IHH), Kallmann syndrome (KS), health-related quality of life (HRQoL)

## Abstract

(1) Background: Isolated hypogonadotropic hypogonadism (IHH) is a genetic condition characterized by impaired puberty and fertility. IHH can significantly impact patient health-related quality of life (HRQoL), sexual satisfaction (SS) and mood. (2) Methods: Participants included 132 IHH subjects (89 men and 43 women) and 132 sex- and age-matched controls. HRQoL, depressive symptoms, erectile dysfunction (ED), and SS were assessed in an online survey using the Zung Self-Rating Depression Scale (SDS), 15D instrument of HRQoL (15D), Sexual Satisfaction Questionnaire (SSQ), and 5-item International Index of Erectile Function (IIEF-5). (3) Results: QoL and SS were significantly lower in the IHH group vs. controls. There was a high rate of ED (53.2% vs. 33%, *p* = 0.008) and depressive symptoms (45.00 ± 17.00 vs. 32.00 ± 12.00, *p* < 0.001) in patients vs. controls. The age of patients at IHH diagnosis inversely correlated with their overall 15D scores. An alarming non-compliance rate was seen (51.6%). No differences were found between scores of patients receiving hormone replacement therapy (HRT) and untreated subjects in any of the scales. (4) Conclusions: The HRQoL, SS, ED, and depression levels observed in IHH patients, despite HRT, are alarming. Late IHH diagnosis may have a particularly negative impact on HRQoL. More attention should be devoted to HRT adherence and various HRQoL aspects of IHH patients.

## 1. Introduction

Isolated hypogonadotropic hypogonadism (IHH), also known as isolated gonadotropin-releasing hormone (GnRH) deficiency or idiopathic hypogonadotropic hypogonadism, is a developmental disorder of the hypothalamic–pituitary axis, causing GnRH deficiency. The lack of pubertal activation of the hypothalamic–pituitary–gonadal axis leads to absent or arrested puberty and infertility [1,2].

IHH, apart from disorders of sexual maturation, may be associated with other congenital anomalies, including olfactory disturbances and disorders of taste and hearing. What is more, skeletal, renal, cardiac, dental, and nervous system defects may also occur [3,4]. IHH can be subdivided into two main types characterized by varying degrees of smell disturbances—the normosmic IHH (nIHH) and Kallmann syndrome (KS). A diminished sense of smell (hyposmia) or its absence (anosmia) is one of the most distinguishing features of KS. At least 10% of IHH develops in adulthood [5]. A spontaneous or drug-induced reversal form of hypogonadism, which can occur in about 10–22% of IHH cases, could be overlooked by patients and physicians [6,7,8]. IHH is a condition that is highly heterogeneous epidemiologically, clinically, and genetically [3,4].

Hypogonadism is associated with a wide range of symptoms in both males and females, including fatigue, loss of libido and orgasmic dysfunctions, increased body fat and sleep disturbances [9]. Erectile dysfunction (ED) is a prevalent problem in IHH males [10,11,12]. It has become increasingly evident that IHH is linked with severe sexual disorders and ED, which may affect physical health and emotional well-being [6,8,11]. Moreover, depression and anxiety have been previously associated with delayed diagnosis and treatment of hypogonadism [10,13]. All of these anomalies may exacerbate psychosexual disturbances in hypogonadal patients. Previous studies have suggested impaired health-related quality of life (HRQoL) in hypogonadism [10,12,14,15,16,17,18]. To date, most studies on HRQoL, ED, and sexual satisfaction (SS) in IHH were conducted on small groups of subjects and concern only young hypogonadal men [10,16,17,18,19].

This study aimed to assess the HRQoL, presence of depressive symptoms, ED, and SS in female and male KS and nIHH subjects from various countries using an online survey. Interactions between these aspects, diagnosis and treatment timing, and adherence to hormonal treatment (HRT) were investigated.

## 2. Materials and Methods

Online support networks were used to recruit 132 patients with self-reported GnRH deficiency (89 men and 43 women; median age 30 ± 21.50 years, range: 18–72 years) to complete an online, self-administered survey in English. Subjects whose clinical data did not match IHH were excluded from the study. Participants were informed about the study aim, their right to withdraw at any point, and that survey completion was equivalent to their informed consent. HRT was defined as testosterone replacement therapy (TRT) or gonadotropin treatment in males and estrogen or oral contraceptive pills/vaginal ring in females. The reversal of hypogonadism was defined as the capability to achieve a normal adult testosterone serum concentration after six months from hormone replacement therapy withdrawal. A control group consisting of 89 healthy males and 43 healthy females, matched for age (group-matching) (median 30.50 ± 17.50, range: 18–71 years), was also recruited to complete the survey. The healthy controls were individuals without reported problems concerning sexual development, maturation, fertility or sense of smell. Four valid and reliable scales were administered to both groups. These included the Zung Self-Rating Depression Scale (SDS), the 15D HRQoL questionnaire (15D), the Sexual Satisfaction Questionnaire (SSQ) and the abridged 5-item International Index of Erectile Function (IIEF-5) [20,21,22,23].

### 2.1. Zung Self-Rating Depression Scale (SDS)

The SDS consists of 20 items, of which 10 are worded positively and 10 negatively, scored on a Likert scale from 1 to 4 [23]. The SDS can be used to assess psychological, affective and physical symptoms of depression. A total score is the sum of scores from each item and ranges from 20 to 80. The majority of respondents with depression have a total score between 50 and 69. Scores above 70 suggest severe depression. The SDS serves as a screening tool and is widely used in clinical as well as research settings to estimate depression severity [23,24]. However, it cannot substitute a comprehensive clinical assessment to establish a diagnosis of the disease [23,24].

### 2.2. 15D

The 15D instrument (Harri Sintonen, Department of Public Health, Helsinki, Finland) is a generic questionnaire designed to measure HRQoL [22]. It consists of 15 different health dimensions—mobility, vision, hearing, breathing, sleeping, eating, speech, excretion, usual activities, mental function, discomfort and symptoms, depression, distress, vitality, and sexual activity. Each dimension is characterized by 1 of 5 ordinal levels to be chosen by the respondent as the level that best relates to their current state of health. The total 15D score is calculated using a valuation algorithm and is a single index number on a 0–1 scale (0 = being dead, 1 = “full” HRQoL, i.e., no problems on any dimension). The minimum important change in the 15D score is 0.015 [25]. Dimension level values are used to create 15D profiles showing the position of the respondent on each of the 15 health dimensions. The 15D score and dimension level values are calculated from the health state descriptive system employing a set of population-based preference or utility weights [22].

### 2.3. Sexual Satisfaction Questionnaire (SSQ)

The SSQ, designed by Nomejko and Dolińska-Zygmunt, is a questionnaire consisting of 10 items scored on a 1–4 Likert scale, which can be used to assess the level of SS [20]. It may serve as a tool to determine the cognitive and emotional approach of the respondents to their own sexual life. Studies have supported the SSQ as a valid and reliable psychometric method. The total score, obtained by a summation of scores for each item, allows for dividing the respondents into subgroups with low, medium or high results. In the case of female participants, the total score was characterized as follows: 10–25 low; 26–31 medium; 32–40 high result. The ranges for males were: 10–27 low; 28–33 medium; 34–40 high result [20,26]. We used an officially available English version provided by the authors.

### 2.4. International Index of Erectile Function-5 (IIEF-5)

The IIEF-5 is an abridged, 5-item version of the IIEF developed to examine the erectile aspect of male sexual function and intercourse satisfaction. It can be utilized in the diagnostics of ED and estimation of its severity, classifying ED into the following five levels depending on the score: 22–25 no ED, 17–21 mild ED, 12–16 mild-to-moderate ED, 8–11 moderate ED, and 5–7 severe ED [21].

Apart from sociodemographic data, participants provided information on their health care experiences, treatment outcomes, and health-related challenges. The study was approved by the Ethics Committee of the Poznan University of Medical Sciences (686/16). The study protocol was in accordance with the ethical guidelines of the Helsinki Declaration developed by the World Medical Association.

### 2.5. Statistical Analysis

Statistical evaluations were performed with Statistica v.13.0 statistical software (StatSoft Polska Sp. z o.o., Krakow, Poland). Statistical significance was set at *p* < 0.05. Based on the obtained data, the number, percentage, mean, median, and standard deviation values were determined. The Shapiro–Wilk test was utilized for the evaluation of the normality of distribution of continuous variables. If these variables were normally distributed, the Student’s *t*-test was used to compare groups, otherwise the Mann–Whitney U test was used. To compare respondents within their groups, if the variables were normally distributed, the t-Student test for paired data was chosen; if not, the Wilcoxon test was used. The chi-square and Fisher’s exact tests were employed to compare discrete variables between independent groups, while the McNemar test was used between dependent groups.

## 3. Results

### 3.1. Patient Characteristics (IHH)

The online survey was completed by 89 males and 43 females with self-reported IHH. The median age of the patients was 30 ± 21.50 years (range: 18–72 years); 23 (17.4%) patients had nIHH, and 109 had KS (82.6%). The studied population was international, as shown in Figure 1. The majority of participants received the IHH diagnosis from an endocrinologist, gynecologist, or andrologist at the age of 10–20 years. Subjects who were married or in a relationship constituted 44.7% of the patient group. Most patients completed higher or vocational education (72.7%). A comparison between sociodemographic data of IHH patients and controls is presented in Table 1. A highly significant difference was seen between patients and controls in the education level. However, in the absence of data on parental socioeconomic status, it is impossible to determine whether the patients’ lower education reflects the impact of the disorder or parental background.

At the time of response, 101 patients (76.5%) were on long-term HRT (>1 year), 6 (4.5%) were on HRT <1 year, and 25 (19%) were untreated. Due to the variety of TRT (69 patients on 15 preparations), gonadotropin, estrogen and birth control preparations, and regimens in the multinational patient group, the analysis of different HRTs was not done. Participants who reported an interruption of hormonal therapy for a period exceeding a year made up 23.5% of the patient group. About half of the patients (51.6% of respondents) reported irregularity in hormonal treatment. The reversal of hypogonadism was reported by eight IHH subjects (6%; seven men, one woman). Up to 41% of IHH patients (*n* = 54) had a history of depression (Table 1).

A comparison between the sociodemographic and clinical data of the nIHH and the KS subjects is shown in Table 1. No statistically significant differences were found between the nIHH and the KS patients based on the analyzed sociodemographic and clinical factors.

### 3.2. Questionnaires

#### 3.2.1. 15D

IHH subjects showed reduced total 15D scores (*p* < 0.001) and scores from each dimension, compared to controls (Table 2, Figure 2). However, the differences found in the 15D total scores and each of the 15D dimensions between KS and nIHH patients were not significant (*p* = 0.34) (Table 3). The median 15D total score and scores from the dimensions of vision, breathing, sleeping, excretion, usual activities, mental function, discomfort and symptoms, depression, distress, vitality, and sexual activity were lower in IHH patients who, based on the SDS, had symptoms of depression (*n* = 44) vs. IHH subjects with no symptoms of depression (*n* = 88) (*p* < 0.05) (Table 4). IHH patients with ED had lower total 15D scores (0.84 ± 0.14 vs. 0.92 ± 0.14; *p* < 0.001) and scores on the dimensions of vision, depression, distress, vitality, and sexual activity (1.0 ± 0.22 vs. 1 ± 0.00, *p* < 0.05; 0.70 ± 0.25 vs. 0.77 ± 0.23, *p* = 0.02; 0.77 ± 0.25 vs. 0.73 ± 0.27, *p* = 0.007; 0.77 ± 0.26 vs. 0.77 ± 0.23, *p* = 0.02; 0.71 ± 0.56 vs. 0.77 ± 0.29, *p* = 0.002), respectively, compared to IHH patients without ED (*p* < 0.05) (Table 4). No significant differences in total 15D scores were found between females and males (0.95 ± 0.11, 0.95 ± 0.10, *p* = 0.81).

#### 3.2.2. SSQ

The SSQ scores were lower in IHH patients vs. healthy controls (26.00 ± 9.00 vs. 33.00 ± 6.00, *p* < 0.001) (Table 2). The SSQ scores of nIHH patients were similar to those of patients with KS (26.00 ± 9.00 vs. 25.00 ± 9.00, *p* = 0.71) (Table 3). Among the SSQ subgroups, a statistically significant difference was found in the frequency and severity of depression symptoms and the presence of ED. An inverse correlation was observed between SSQ scores and the occurrence of depressive symptoms and ED (*p* < 0.05).

#### 3.2.3. IIEF-5

Male IHH subjects had a higher prevalence of ED vs. controls (53.2% vs. 33%, *p* = 0.008), as assessed by the IIEF-5. When the cut-off value of the IIEF-5 score was accepted as 22 points, 55.6% of the untreated patients with IHH experienced ED vs. 53% of the treated IHH subjects (*p* = 0.88). Patients with ED had lower median scores of SSQ vs. IHH patients without ED (24.00 ± 7.00 vs. 30.00 ± 5.5, *p* < 0.001) (Table 4). A higher prevalence of depressive symptoms was present in IHH subjects with ED compared to those without ED (68.1% vs. 31.2%, *p* = 0.001) (Table 4). Higher SDS scores were observed in IHH subjects with ED vs. those without ED (45.00 ± 13.00 vs. 37.00 ± 17.00, *p* = 0.002).

#### 3.2.4. SDS

Patients with IHH had significantly higher raw total scores for SDS than the control subjects (45.00 ± 17.00 vs. 32.00 ± 12.00, *p* < 0.001). With a cut-off value of 50 for the SDS index, 33.3% of the IHH patient group presented with depressive symptoms, whereas this rate was only 11.3% in the control group (*p* < 0.001). Only 34.1% of IHH patients with an SDS index above 50 were taking antidepressants. No significant differences were found in the SDS index scores between men and women (41.25 ± 17.5, 40.0 ± 13.8; *p* = 0.84).

### 3.3. Treatment, Satisfaction with Therapy, and Adherence

The differences found between the 15D, SDS, SSQ, and IIEF-5 scores were not statistically significant between the patients treated with HRT for at least one year and the untreated subjects (Table 4). Patients satisfied with HRT (*n* = 75; 70.8%) had higher median total 15D scores and scores on the dimensions of sleeping, depression, distress, vitality, and sexual activity (0.87 ± 0.14 vs. 0.80 ± 0.16, *p* < 0.001; 0.79 ± 0.49 vs. 0.51 ± 0.46, *p* = 0.003; 0.73 ± 0.48 vs. 0.48 ± 0.46, *p* = 0.006; 0.77 ± 0.48 vs. 0.51 ± 0.26, *p* = 0.017; 0.71 ± 0.56 vs. 0.44 ± 0.75, *p* = 0.03), as well as higher SSQ scores (26.00 ± 9.00 vs. 22.00 ± 10.00, *p* = 0.003, respectively) vs. unsatisfied IHH subjects (*n* = 31; 29.2%) (Table 4). The SDS raw scores were higher in IHH subjects unsatisfied with HRT vs. patients satisfied with HRT (51.00 ± 11.00 vs. 41.00 ± 18.00, *p* < 0.001).

Patients who declared having irregularities in their HRT had mean overall 15D, SSQ, and SDS scores that were similar to those receiving HRT regularly (Table 4).

### 3.4. Reversal Form of IHH

The total 15D scores, scores in all symptom domains, and the overall SSQ, SDS, and IIEF-5 scores did not differ between patients with or without a reported reversal form of IHH.

### 3.5. Correlation Analysis

A negative correlation was found between the age of patients at the time of IHH diagnosis and their overall 15D score, as well as the scores for: mobility, vision, breathing, discomfort and symptoms (r = −0.190, *p* = 0.03; r = −0.243, *p* = 0.004; r = −0.237, *p* = 0.006, r = −0.212, *p* = 0.01, respectively). No statistically significant association was found between patient age at the time of the study and the overall scores from any of the questionnaires used. The duration of hormonal treatment positively correlated with the SSQ score (r = 0.241, *p* = 0.01) and the following 15D dimensions: eating, speech, distress, vitality, and sexual activity (r = 0.270, *p* = 0.003; r = 0.181, *p* = 0.049; r = 0.249, *p* = 0.006; r = 0.186, *p* = 0.043; r = 0.194, *p* = 0.035), and negatively with the SDS score (r = −0.256; *p* = 0.005). The overall 15D score was positively associated with the SSQ (r = 0.395, *p* < 0.001) and IIEF-5 scores (r = 0.285, *p* < 0.05) and negatively with the SDS score (r = −0.785, *p* < 0.001). A positive correlation was also observed between the SSQ score and the scores on the 15D dimensions of breathing, sleeping, usual activities, depression, distress, vitality, and sexual activity (r = 0.311; *p* = 0.006; r = 0.252; *p* = 0.026; r = 0.268; *p* = 0.018; r = 0.437; *p* < 0.001; r = 0.339; *p* < 0.003; r = 0.338; *p* = 0.002; r = 0.605; *p* < 0.001, respectively). The SSQ scores positively correlated with the IIEF-5 scores (r = 0.556; *p* < 0.001).

## 4. Discussion

### 4.1. Health-Related Quality of Life

This study showed that IHH is associated with a lower HRQoL compared to healthy subjects, confirming previous observations (*p* < 0.001). Patients with IHH also reported worse QoL in all dimensions assessed in 15D. Levels of HRQoL appear to be comparable in nIHH and KS (*p* > 0.05). The presence of dysosmia does not seem to have an additional negative impact on HRQoL in KS. However, the majority of patients evaluated in our study had dysosmia (82.6%). Such a significant predominance of KS may limit the conclusions about nIHH that could be drawn from this research and may result from the dominating participation of the “Kallmann Syndromers” Facebook group members in the online survey. Unfortunately, there seems to be a lack of data comparing the QoL of KS and nIHH patients in the current literature.

The levels of HRQoL seem to be comparable between the IHH females and males we assessed. Although the predominance of male vs. female patients was not as prominent in this study (67.4% vs. 32.6%), it is important to remember that there are considerable differences in IHH incidence between sexes, with males being diagnosed three to five times more often than females [4,27,28]. Such differences in IHH prevalence contribute to the fact that many crucial aspects of patient management, such as the HRQoL, do not receive equal research attention in both sexes. Most of the available data from previous studies on QoL in IHH concern the male population, limiting result comparisons between male and female patients.

The HRQoL of the international IHH patient group we assessed appears to be affected on a greater number of levels than the Finnish male IHH population studied by Varimo et al. [18]. In that study, IHH patients demonstrated impairment in only two 15D dimensions (depression and distress) [18]. In the research by Georgopoulos et al., most aspects of HRQoL measured with WHOQOL-BREF were similar in subjects with GnRH deficiency on HRT to those of healthy controls, except for the lower satisfaction with general health in patients compared to controls (*p* < 0.001) [29]. In that study, the predominance of nIHH over KS cases was significant in contrast to the population evaluated in the present study. Such differences in findings regarding HRQoL could result from small sample sizes, population characteristics, and possible variations in the quality of IHH treatment received by participants.

Research suggests that delayed diagnosis and management of hypogonadism may be related to adverse psychological outcomes. As the current study and previous research indicate, a delay in diagnosis and treatment may deteriorate patient HRQoL (Table 4) [11]. An inverse relationship between the age at IHH diagnosis and the overall 15D score was observed in both the present study and the research by Varimo et al., emphasizing the importance of an early diagnosis in the efforts to support a satisfactory level of HRQoL among IHH individuals [18].

### 4.2. Sexual Satisfaction

IHH patients suffer from multi-causal and multidimensional sexual dysfunction [11,12,15,30,31]. Sexual activity seems to be one of the main concerns for subjects with hypogonadism [12,32]. Compared with a large survey of US adults in which 5.4% of men reported never being sexually active, as many as 26% of IHH men reported sexual inactivity in another study (*p* < 0.001) [12,33]. In female hypogonadism, only 50–80% of patients declare being sexually active [34,35]. In the current study, 94% of IHH patients responded to SSQ regarding satisfaction from their sexual life and thus declared some level of sexual activity.

The results from this study show that SS was significantly impaired in males and females with IHH compared to controls (*p* < 0.001). Up to 52.3% of IHH patients had low SSQ results vs. 6.1% of controls (*p* < 0.001). Comparable SS was seen in female and male IHH subjects, and there were no statistically significant differences in this aspect between patients aged below 35 and those above 35 years. Additionally, no differences were found in SS between nIHH and KS patients (*p* < 0.05). Of note, the length of hormonal treatment was a factor positively influencing the SS of IHH patients. Due to the different norms for males and females in the SSQ, it could not be used to compare sexes.

Contrary to our findings, research by Georgopoulos et al. suggested that the overall SS was similar in women with GnRH deficiency on HRT to those of healthy controls, based on the Female Sexual Function Index results. Similarly, the SS of GnRH-deficient males under treatment was comparable to that of healthy men, as measured by the 15-item IIEF. Sexual desire, however, was found to be significantly lower in GnRH-deficient females on HRT than in controls [29]. In the current study, SS measured with the SSQ score was highly correlated with the “sexual activity” dimension score in the 15D instrument and with the total 15D score reflecting overall HRQoL (r = 0.605; *p* < 0.001; r = 0.394, *p* < 0.001). Although sexuality is an important aspect of QoL in IHH, little research concerning the impact of SS on QoL in IHH has been conducted thus far [29,36].

Sexual dissatisfaction could be linked to the late introduction of HRT or non-optimal HRT. More research on the relationship between different treatment regimens and not only fertility, but also the SS, of patients with hypogonadism are needed in the future.

### 4.3. Erectile Dysfunction

A high incidence of ED in IHH men, at comparable levels in KS and nIHH, was observed in the present study. According to the IIEF-5 results, as much as 53.2% of IHH males reported ED. Research suggests that GnRH-deficient males on HRT may experience minor ED and poorer orgasmic experiences; however, adjustments for anxiety and depression reduce such differences to insignificant levels [17,29]. In our study, patients with ED were worse off not only on the 15D dimension of sexual activity, but also of vision, depression, distress and vitality, and reported lower SS (Table 4). Moreover, a positive association between the IIEF-5 and the overall 15D scores was evident in the current study (r = 0.285, *p* < 0.05). Ferlin et al. studied sexual dysfunction in Klinefelter syndrome patients employing the IIEF and suggested that ED is primarily related to psychological disturbances, whereas sexual desire, intercourse satisfaction, and overall SS can be linked to the hypogonadal state and decreased testosterone levels [36]. Although ED and depression are frequent comorbidities, causality (which could be bidirectional) has not been established [37,38]. In IHH, psychological issues, the presence of microphallus, or a history of cryptorchidism could markedly impact the prevalence of sexual difficulties and ED [12,27,30]. Unfortunately, no reliable data on early clinical signs of severe GnRH deficiency (such as microphallus or cryptorchidism) or additional congenital anomalies were obtained in the present study. When it comes to hormonal treatment, it does not appear to reverse ED fully or reduce its rates. A similarly high incidence of ED was found in treated and untreated IHH patients (53% vs. 55.6%, *p* = 0.88), compared to ED in a third of male controls in this study. A small but statistically significant increase in erectile function in a multinational population of middle-aged and older hypogonadal men on TRT was shown by Rosen et al. [31]. In contrast to our study, Georgopoulos et al. noted that GnRH-deficient males on HRT, with testosterone values within the normal range, reported similar erectile function in IIEF as healthy, age-matched individuals when results were adjusted for anxiety and depression (*p* = 0.127) [29]. Further studies focused on the influence of different HRT regimens on erectile function in large groups of IHH patients are necessary to study this important aspect more extensively.

### 4.4. Depression

Hypogonadal patients suffer from depressive symptoms more often than the healthy population [10,16,30]. In this study, almost 41% of enrolled IHH patients had a previous history of depression and one-third of IHH patients had symptoms of depression at the time of the study. Depressive symptoms were present just as often in KS as in nIHH, independently of dysosmia, and occurred with a similar frequency in males and females, regardless of age.

Similarly to the rate of depressive symptoms among patients in the current study (33%), psychiatric disorders such as depression, bipolar disorder, or anxiety disorder were seen in 27% of IHH patients described by Varimo et al. [18]. An even higher prevalence was found by Dzemaili et al., who observed that more than half of IHH women (receiving HRT for at least a year) report depressive symptoms, as assessed with SDS; IHH men may experience comparably alarming depression rates [14,39,40]. The high incidence of depressive symptoms experienced by IHH subjects in our study, both in the past and at the time of participation, may reflect bias in sampling. Patients involved in online support groups may not represent the IHH population, including patients most bothered by the disease or its treatments. We observed that only a third of patients with symptoms suggesting depression receive antidepressant medications, highlighting the need to address this problem in the course of IHH treatment [41].

### 4.5. Hormonal Treatment Satisfaction and Adherence

IHH patients typically require lifelong hormonal treatment [2]. Surprisingly, we found that the untreated patients experienced a comparable level of overall HRQoL, SS and depressive symptoms as the hormonally treated IHH subjects. Our results are contrary to those of Mileski et al., who observed that QoL is similar in hypogonadism under regular TRT compared to healthy male volunteers [42]. The fact that a relatively small group of untreated patients completed the given questionnaire could have influenced such a comparison in our study. It may be the case that the patients untreated at the time of the study had less severe hypogonadism or experienced a reversal of hypogonadism (which can occur in about 10–22% of IHH cases) [7,8]. The current results show that long-term HRT (>1 year) appears to positively influence the level of SS, the presence of depressive symptoms, and the 15D dimensions of eating, speech, depression, distress, vitality, and sexual activity, but not the overall HRQoL.

When it comes to satisfaction with HRT, it seems to have a positive impact on HRQoL, especially on the dimensions of sleeping, usual activities, discomfort, depression, and distress. Our results confirmed previous observations regarding SS as an important determinant of satisfaction with HRT [11,14]. However, poor HRT adherence was observed in almost half of the patients. Low adherence to treatment was even more prevalent (62%) among IHH patients in a study by Dzemaili et al. [14]. HRT adherence was linked to better erectile function in the present study. Surprisingly, the regularity of treatment did not appear to impact HRQoL, SS, or the presence of depressive symptoms. A possible reason for such findings may be that patients with a less severe reproductive phenotype may not experience adverse psychological or sexual symptoms due to minor treatment irregularities. A gap in treatment exceeding one year was reported by 23.5% of IHH patients, compared with 37–55% stated in previous reports [11,14]. IHH is a chronic disease, and life-long treatment adherence could be challenging. The high risk of ED in both treated and untreated IHH males could also influence HRT adherence. Considering the low perceived QoL and SS of patients, the IHH therapeutic approach may be inefficient in providing adequate improvement in these areas.

### 4.6. Reversal of Hypogonadism

Interestingly, the experience of hypogonadism reversal does not appear to affect the HRQoL, SS and ED. Comparable findings on HRQoL were obtained by Varimo et al. [18]. The reversal of hypogonadism does not necessarily influence the presence of other potential congenital anomalies or additional disabilities, which could also affect the HRQoL of patients.

### 4.7. Limitations

The main limitations of this study were the self-categorization of patients and biased sampling of individuals. Online support groups may include disproportionately distressed patients, and controlling for fake data is difficult. Another disadvantage of this study was that no data on patient sex hormone levels were available. Although the use of IIEF-5 reduced the questionnaire length, making it easier for respondents to complete, to better evaluate sexual function in men, the full, 15-item IIEF could have been used instead. In further studies, tools such as the Morisky Medication Adherence Scale could be applied to define the concept of irregular HRT more precisely. The studied population was multinational, but the sample size was too small to compare the different nationalities and HRT preparations and regimens. Patients and controls were not matched for nationality, ethnicity, race, relationship status, and parental socioeconomic status. Despite the limitations, the advantage of this study is that it involved a relatively large patient group, which is uncommon in IHH research due to the rarity of this condition. In addition, a variety of questionnaires were used to examine the broad concept of the QoL in detail.

## 5. Conclusions

This study highlights the severe, multidimensional impairment of HRQoL and SS, and the high occurrence of depressive states in IHH patients. The findings proved that HRQoL is inextricably linked to SS and mental health in IHH subjects, not only to hypogonadism itself. Late diagnosis and delays in treatment of IHH may have a particularly negative effect on the HRQoL. The high level of depression and non-compliance with treatment is alarming. Both treated and untreated IHH males are at increased risk of having ED. SS should be an important treatment goal, being one of the key components of patient wellbeing. Consideration should be given to using tools to assess HRQoL in all IHH individuals at diagnosis and regular follow-up, as lower HRQoL scores could indicate patients at risk of depression or sexual problems. Further research is needed to identify whether early, intensive and individualized hormonal treatment could improve the HRQoL and the sexual and psychological functioning of IHH patients.

## Figures and Tables

**Figure 1 jcm-10-02622-f001:**
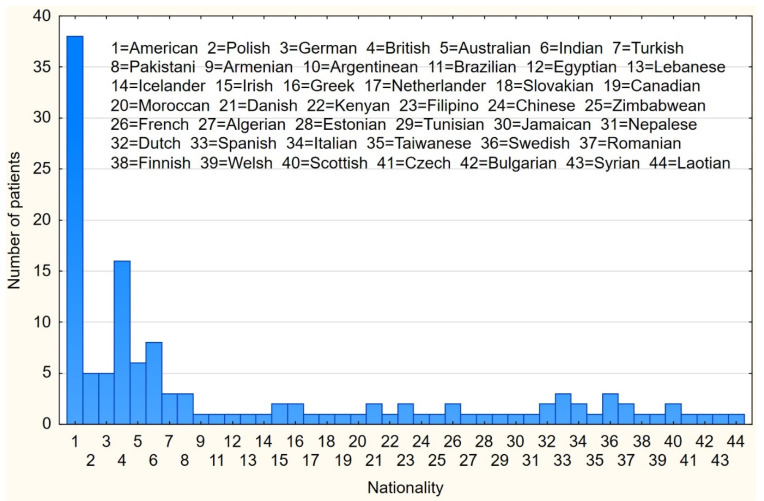
Nationality of IHH patients.

**Figure 2 jcm-10-02622-f002:**
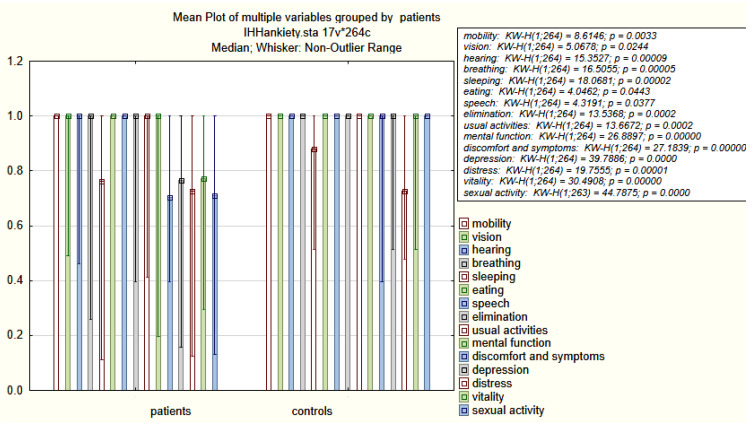
Median (IQR) values of 15D dimensions in IHH patients and the control group.

**Table 1 jcm-10-02622-t001:** Sociodemographic data and characteristics of IHH patients (*n* = 132) and control subjects (*n* = 132).

	Total IHH (*n* = 132)	KS (*n* = 109)	nIHH (*n* = 23)	*p*-Value (KS vs. nIHH)	Controls (*n* = 132)	*p*-Value (Total IHH vs. Controls)
Median ± IQR						
age in years	30 ± 21.50	31 ± 17	30 ± 17	NS	30.50 ± 17.50	NS
Age	*n* (%)	*n* (%)	*n* (%)		*n* (%)	
18–29	63 (47.7%)	52 (47.7%)	11 (47.9%)		64 (48.5%)	
30–39	29 (22%)	25 (22.9%)	4 (17.4%)		29 (22%)	
40–49	20 (15.2%)	15 (13.8%)	5 (21.7%)		19 (14.4%)	
50–59	11 (8.3%)	9 (8.3%)	2 (8.7%)		12 (9.1%)	
60+	9 (6.8%)	8 (7.3%)	1 (4.3%)		8 (6%)	
Sex						
male	89 (67.4%)	72 (66.1%)	17 (73.9%)	NS	89 (67.4%)	NS
female	43 (32.6%)	37 (33.9%)	6 (26.1%)		43 (32.6%)	
Education						
primary	8 (6.1%)	7 (6.4%)	1 (4.3%)		0 (0%)	
secondary	28 (21.2%)	23 (21.1%)	5 (21.7%)	NS	12 (9.1%)	<0.001
higher/vocational	96 (72.7%)	79 (72.5%)	17 (74%)		120 (90.9%)	
Employment						
student	11 (8.5%)	10 (9.4%)	1 (4.35%)		62 (47.3%)	
unemployed	7 (5.4%)	5 (4.7%)	2 (8.7%)		0 (0%)	
worker retired	97 (75.2%) 7 (5.4%)	80 (75.5%) 6 (5.7%)	17 (73.9%) 1 (4.35%)	NS	68 (51.9%) 1 (0.8%)	<0.001
stays at home	5 (3.9%)	3 (2.8%)	2 (8.7%)		0 (0%)	
disabled	2 (1.6%)	2 (1.9%)	0 (0%)		0 (0%)	
Relationship status						
single	68 (51.5%)	56 (51.4%)	12 (52.2%)		84 (63.6%)	
married	50 (37.9%)	42 (38.5%)	8 (34.8%)		26 (19.7%)	
divorced	5 (3.8%)	5 (4.6%)	0 (0%)		2 (1.52%)	
partner	9 (6.8%)	6 (5.5%)	3 (13%)	NS	19 (14.4%)	0.004
relationship						
widow	0 (0%)	0 (0%)	0 (0%)		1 (0.8%)	
Age at diagnosis						
1–10	10 (7.6%	9 (8.3%)	1 (4.3%)			
11–20	73 (55.3%)	61 (56%)	12 (52.2%)			
21–30	41 (31.1%)	32 (29.3%)	9 (39.2%)	NS	N/A	N/A
31–40	6 (4.5%)	6 (5.5%)	0 (0%)			
41–50	2 (1.5%)	1 (0.9%)	1 (4.3%)			
51–60	0 (0%)	0 (0%)	0 (0%)			
60+	0 (0%)	0 (0%)	0 (0%)			
History of depression						
Yes	54 (40.9%)	43 (39.4%)	11 (47.8%)	NS	19 (14.4%)	<0.001
No	78 (59.1%)	66 (60.6%)	12 (52.2%)		113 (85.6%)	
Reversal of IHH						
Yes No	8 (6.1%) 124 (93.9%)	7 (6.4%) 102 (93.6%)	1 (4.3%) 22 (95.7%)	NS		
HRT/fertility treatment						
Yes	107 (81.1%) 25 (18.9%)	89 (81.6%) 20 (18.4%)	18 (78.3%) 5 (21.7%)	NS		
No						
Length of treatment						
<one year	12 (10.3%)	11 (11.6%)	1 (4.3%)			
1–5 years 5–10 years	33 (28.5%) 15 (12.9%)	26 (27.4%) 13 (13.7%)	7 (30.4%) 2 (8.7%)	NS		
>10 years	56 (48.3%)	45 (47.3%)	11 (47.8%)			
Treatment irregularity *						
Yes No	63 (51.6%) 59 (48.4%)	56 (55.5%) 45 (44.5%)	7 (33.3%) 14 (76.7%)	NS		
Satisfaction with hormonal treatment						
Yes No	84 (64.1%) 47 (35.9%)	73 (67.6%) 35 (32.4%)	11 (47.8%) 12 (52.2%)	NS		

HRT—hormone-replacement therapy; IHH—isolated hypogonadotropic hypogonadism; KS—Kallmann syndrome; nIHH—normosmic IHH; * Treatment irregularity was defined as any time when a patient missed a prescribed dose of treatment for at least 2 days in a period of more than 2 consecutive months.

**Table 2 jcm-10-02622-t002:** Scores from the 15D instrument (15D), the Zung Self-Rating Depression Scale (SDS), the Sexual Satisfaction Questionnaire (SSQ), and the 5-item International Index of Erectile Function (IIEF-5)—data from patients with isolated hypogonadotropic hypogonadism (IHH) and controls.

15D				SDS			
	IHH Patients Median (IQR)	Controls Median (IQR)	*p*-Value		IHH Patients Median (IQR)	Controls Median (IQR)	*p*-Value
15D score	0.85 ± 0.16	0.95 ± 0.10	<0.001	SDS score	45.00 ± 17.00	32.00 ± 12.00	<0.001
Mobility	1.00 ± 0.00	1.00 ± 0.00	0.003	SDS index	56.25 ± 21.25	40.00 ± 15.00	<0.001
Vision	1.00 ± 0.22	1.00 ± 0.00	0.024	Normal range *	88 (66.7%)	117 (88.6%)	<0.001
Hearing	1.00 ± 0.25	1.00 ± 0.00	<0.001	Mildly depressed *	32 (24.2%)	11 (8.3%)	<0.001
Breathing	1.00 ± 0.30	1.00 ± 0.00	<0.001	Moderately depressed *	11 (8.3%)	4 (3%)	NS
Sleeping	0.76 ± 0.49	0.88 ± 0.24	<0.001	Severely depressed *	1 (0.8%)	0 (0%)	NS
Eating	1.00 ± 0.00	1.00 ± 0.00	0.045	SSQ			
Speech	1.00 ± 0.00	1.00 ± 0.00	0.038	SSQ score	26.00 ± 9.00	33.00 ± 6.00	<0.001
Excretion	1.00 ± 0.32	1.00 ± 0.00	<0.001	Low results **	69 (55.6%)	8 (6.9%)	<0.001
Usual activities	1.00 ± 0.28	1.00 ± 0.00	<0.001	Medium results **	38 (30.65%)	41 (35.3%)	0.69/>0.05/NS
Mental function	1.00 ± 0.36	1.00 ± 0.00	<0.001	High results **	17 (13.7%)	67 (57.8%)	<0.001
Discomfort and symptoms	0.70 ± 0.30	1.00 ± 0.30	<0.001	IIEF-5			
Depression	0.77 ± 0.25	1.00 ± 0.23	<0.001	IIEF-5 score	21.00 ± 8.00	24.00 ± 4.500	<0.001
Distress	0.73 ± 0.25	0.73 ± 0.27	<0.001				
Vitality	0.77 ± 0.26	1.00 ± 0.23	<0.001				
Sexual activity	0.71 ± 0.56	1.00 ± 0.00	<0.001				

SDS—Zung Self-Rating Depression Scale; * number of patients in the following SDS score ranges: 25–49—normal range, 50–59—mildly depressed, 60–69—moderately depressed, 70 and above—severely depressed. SSQ—Sexual Satisfaction Questionnaire; ** number of patients in the following SSQ score ranges: women—10–25 low result, 26–31 medium result, 32–40 high result; men—10–27 low result, 28–33 medium result, 34–40 high result. 15D, SDS and SSQ scores and related percentages refer to subjects who completed the given questionnaire. The 15D data of patients and controls were compared using the Mann–Whitney U test. Data are presented as median ± IQR.

**Table 3 jcm-10-02622-t003:** Scores from the 15D instrument (15D), the Zung Self-Rating Depression Scale (SDS), the Sexual Satisfaction Questionnaire (SSQ), and the 5-item International Index of Erectile Function (IIEF-5)—data from Kallmann syndrome (KS) and normosmic IHH (nIHH) patients.

15D					SDS				
	KS Patients Median (IQR)	nIHH Patients Median (IQR)	*p*-Value KS vs. nIHH	*p*-Value KS vs. Controls		KS Patients Median (IQR)	nIHH Patients Median (IQR)	*p*-Value KS vs. nIHH	*p*-Value KS vs. Controls
15D score	0.85 ± 0.16	0.87 ± 0.18	0.34	<0.001	SDS score	45.00 ± 17.00	45.00 ± 18.00	0.59	<0.001
Mobility	1.00 ± 0.00	1.00 ± 0.00	0.53	0.002	SDS index	56.25 ± 21.25	56.25 ± 22.50	0.59	<0.001
Vision	1.00 ± 0.22	1.00 ± 0.00	0.28	0.009	Normal range *	71 (67.1%)	14 (60.9%)	0.58	<0.001
Hearing	1.00 ± 0.25	1.00 ± 0.00	0.45	<0.001	Mildly depressed *	26 (24.5%)	6 (26.1%)	0.88	<0.001
Breathing	1.00 ± 0.30	1.00 ± 0.00	0.06	<0.001	Moderately depressed *	8 (7.5%)	3 (13%)	0.42	0.11
Sleeping	0.76 ± 0.49	0.76 ± 0.49	0.31	<0.001	Severely depressed *	1 (0.9%)	0 (0%)	0.64	0.26
Eating	1.00 ± 0.00	1.00 ± 0.00	0.36	0.025	SSQ				
Speech	1.00 ± 0.00	1.00 ± 0.00	0.19	0.030	SSQ score	26.00 ± 9.00	25.00 ± 9.00	0.71	<0.001
Excretion	1.00 ± 0.32	1.00 ± 0.00	0.12	<0.001	Low results **	55 (56.1%)	12 (52.2%)	0.98	<0.001
Usual activities	1.00 ± 0.28	1.00 ± 0.28	0.88	<0.001	Medium results **	29 (29.6%)	8 (34.8%)	0.48	0.53
Mental function	1.00 ± 0.36	1.00 ± 0.36	0.70	<0.001	High results **	14 (14.3%)	3 (13%)	0.98	<0.001
Discomfort and symptoms	0.70 ± 0.30	0.70 ± 0.30	0.91	<0.001	IIEF-5				
Depression	0.77 ± 0.25	0.77 ± 0.49	0.66	<0.001	IIEF-5 score	21.00 ± 8.00	19.00 ± 8.00	0.09	0.005
Distress	0.73 ± 0.25	0.73 ± 0.52	0.36	<0.001					
Vitality	0.77 ± 0.26	0.77 ± 0.26	0.90	<0.001					
Sexual activity	0.71 ± 0.56	0.71 ± 0.56	0.45	<0.001					

nIHH—normosmic isolated hypogonadotropic hypogonadism; KS—Kallmann syndrome; SDS—Zung Self-Rating Depression Scale; * number of patients in the following SDS score ranges: 25–49—normal range, 50–59—mildly depressed, 60–69—moderately depressed, 70 and above—severely depressed. SSQ—Sexual Satisfaction Questionnaire; ** number of patients in the following SSQ score ranges: women—10–25 low result, 26–31 medium result, 32–40 high result; men—10–27 low result, 28–33 medium result, 34–40 high result. 15D, SDS and SSQ scores and related percentages refer to subjects who completed the given questionnaire. The 15D data of patients and controls were compared using the Mann–Whitney U test. Data are presented as median ± IQR.

**Table 4 jcm-10-02622-t004:** Comparison between IHH patient scores from 15D and SSQ in relation to SDS index scores, IIEF-5 scores, age at diagnosis, HRT received, treatment satisfaction and adherence.

		15D Score		SSQ	
		Median (IQR)	*p*-Value	Median (IQR)	*p*-Value
Depressive symptoms *					
Absent (*n* = 44) Present (*n* = 88)		0.94 ± 0.08 0.80 ± 0.14	<0.001	31.00 ± 7.00 24.00 ± 7.00	<0.001
ED	Absent Present	0.92 ± 0.14 0.84 ± 0.14	<0.001	30.00 ± 5.50 24.00 ± 7.00	<0.001
Age at diagnosis					
<20		0.86 ± 0.13	<0.05	26.00 ± 9.00	NS
>20		0.82 ± 0.15		25.00 ± 10.00	
HRT					
Yes (*n* = 107) No (*n* = 25)		0.85 ± 0.17 0.86 ± 0.12	NS	25.00 ± 9.50 27 ± 6.50	NS
Satisfaction with HRT					
Yes (*n* = 75)		0.87 ± 0.15	<0.001	26.00 ± 9.00	0.003
No (*n* = 31)		0.80 ± 0.16		22.00 ± 10.00	
Treatment irregularity **					
Yes (*n* = 63) No (*n* = 59)		0.84 ± 0.15 0.87 ± 0.17	0.20	25.00 ± 11.00 25.50 ± 10.00	0.40

* SDS index > 50.** Treatment irregularity was defined as any time when a patient missed a prescribed dose of treatment for at least 2 days in a period of more than 2 consecutive months. 15D—15D instrument; ED—erectile dysfunction; HRT—hormone replacement therapy; SDS—Zung Self-Rating Depression Scale; SSQ—Sexual Satisfaction Questionnaire.

## Data Availability

The data that support the findings of this study are available from the corresponding author, M.K., upon reasonable request.

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
