# Peer review of "Reduced Quality of Life and Sexual Satisfaction in Isolated Hypogonadotropic Hypogonadism"

_jcm, 2021, doi:10.3390/jcm10122622_

Round 1

Reviewer 1 Report

The MS from Kaluzna M and coworkers is an interesting paper focusing on the sexual function of patients with congenital hypogonadotropic hypogonadism. This is often a neglected subject in these patients with a more intensive literature focused on the pathogenesis and clinical management of this rare endocrine disease. Although there are some limitations, however indicated and recognized by the Authors, the study is well designed and presented.

I have only minor observations:

  • Introduction, line 40: consider citing actual ref 27 from Boehm U et al here
  • Results line 143: check the sentence “the median age of…..”
  • Discussion paragraph 4.2: check the sentence “IHH men are even five times more sexual inactivity…”. It should be rewritten ad “IHH men are five times more sexually inactive…”

Author Response

Thank you very much for all your comments and suggestions. We truly believe this is an important topic that is often not fully recognized. The reluctance of patients to stick to HRT points to the importance of comprehensive patient education on the influence of treatment on disease symptoms and regular monitoring of the quality of life and sexuality with regard to the given treatment. Internet survey studies do have their limitations. However, they enable a wider recognition of the condition and facilitate research planning on a larger patient population with complete clinical data and cooperation with international specialists. Following your advice, we have added the citation and changed the marked sentences in the Results and Discussion sections. Once again, thank you for your detailed and accurate review.

Reviewer 2 Report

The authors compared parameters related to overall, health and sexual quality of life in a mixed group of 139 women and men with isolated hypogonadotropic hypogonadism (IHH) gathered from online-support groups, and an age-matched controls. The data comprised of self-rated reported QOL and sexual satisfaction scales. The authors conclude that  the various parameters of QOL were adversely affected in the IHH group. Another  interesting observation is the reluctance of the patients to stick to HRT. The study seems to have some methodological problems (most of which acknowledged by the authors themselves), and it  adds little to the available knowledge on the subject matter. It might be preferable if the authors chose to limit the scope of the study, e.g. to use of HRT snd compliance to treatment.

  1. Abstract and Introduction: The authors refer to isolated hypogonadotropic hypogonadism (IHH) as a "developmental" or "genetic condition". Overall, the genetic forms of IHH are probably the minority of IHH cases in the adult population, unless the authors refer to congenital IHH, which is not clear from the manuscript.
  2. Methods (line 76): Since this was an international survey, how were the controls matched for nationality/ethnic/cultural diversities? Also, the patient' groups combined both genders. It would have been desirable to separate those groups or better, analyst the data by multivariate analysis.
  3. Table 1: The particularly high rate of students in the control group somehow may suggests that that the controls were recruited from a convenient academic environment (?). Controls should have been matched on more parameters than age alone.
  4. Table 1: It seems strange to me that a larger proportion of IHH patients were married compared with the controls (see #3).
  5. I wonder if the negative results in the comparison between the relatively large group of KS and small number of nIHH are valid. It could be preferable to limit the study to KS alone. 
  6. The discussion should be shortened and streamlined.
  7. Discussion P.16, before last paragraph: The sentence starting "compared with large (n>2000)..." is not clear.

Author Response

Dear Reviewer, thank you for all the valuable and accurate suggestions. Our study does have some limitations because it was carried out online and so the survey could not be too long; otherwise, there was a risk that participants would not be willing to complete it. Therefore, unfortunately, the questionnaire could not cover all possible issues. Nevertheless, we believe this study is valuable since it has highlighted several important problems among this patient group, e.g., poor treatment compliance, reluctance to follow HRT regimens and the fact that many patients seem to interrupt their treatment without physician guidance. Moreover, we have recognized the potential lack of proper patient education regarding the importance of HRT, the possibility of insufficient doctor-patient communication and the lack of routine monitoring of mood, depressive symptoms, and sexual life satisfaction in this patient group. Before the collection and the statistical analysis of questionnaire results, we did not expect that the unwillingness to stick to HRT would be one of the main issues among patients since we did not observe such trends in our clinical experience with Polish patients. We are planning to research the use of HRT and treatment compliance further; however, without this initial study and the publication of its results, we will be unable to continue our research. We are planning to conduct studies on more homogenous groups where treatment methods would be comparable.

  1. Thank you for the important comment. According to UpToDate, the condition is described as isolated gonadotropin-releasing hormone deficiency (or idiopathic hypogonadotropic hypogonadism), and it can be subdivided into the normosmic type and the type with olfactory disturbances, the Kallmann syndrome. Based on the information from UpToDate and the renowned IHH specialist, Nelly Pitteloud, M.D., "Some experts consider this to be a congenital disorder and refer to it in the literature as 'congenital hypogonadotropic hypogonadism'. However, more often, these neonatal features are absent and the age of onset or its precise etiology cannot be determined, and thus, the term 'idiopathic' is used.". Only 50-60% of cases of isolated hypogonadotropic hypogonadism have a proven genetic background. Around 10% of cases are of adult onset and so it is not recommended to use the term "congenital". Literature on IHH commonly uses terms such as "developmental" or "genetic condition" and underlines the specific reference to the isolated hypogonadotropic-releasing hormone deficiency (idiopathic hypogonadotropic hypogonadism).
  2. The controls were not matched with patients regarding nationality, ethnicity or cultural background. Unfortunately, this was technically impossible with a group of patients coming from 44 different countries, and we have stressed that in the Limitations section: “Patients and controls were not matched for nationality, ethnicity, race, relationship status and parental socioeconomic status". Further research should involve patients of selected nationalities, which would be selected as the most common ones among respondents, and these would then be matched with controls.

The study included 43 female patients with nIHH/KS. We have described the comparison of survey results between males and females in the main text instead of tables in order not to duplicate data. There were no significant differences in the 15D and SDS scores between males and females. As we emphasized in the discussion, due to different norms for males and females in the SSQ, it could not be used to compare sexes.

We decided to subdivide participants into normosmic and anosmic groups, and this division is presented in Table 2b. We believe that additional divisions in the same study could blur the overall picture. This is, however, one of the major problems with studies on IHH, which affects men much more commonly than women. Therefore, further studies involving nIHH and KS women only are certainly needed. In the current study, basic static methods have been selected, and the use of methods such as multivariate analysis is planned in the continuation of this study. We are planning also further research on treatment adherence on separate groups of nIHH/KS men and women.

  1. Controls were recruited in the same (online) manner among international groups, e.g., language exchange clubs. Age- and gender-matching was the most important for us. As we stressed in the Limitations and the point above, "Patients and controls were not matched for nationality, ethnicity, race, and parental socioeconomic status". It seems that students could be a group most willing to complete surveys on sexual life and quality of life. The questionnaire included a number of verification questions. The control group was selected as precisely as possible and included individuals without reported problems concerning sexual development, maturation, fertility, or sense of smell. These conditions led to the fact that many responses had to be excluded.
  2. Thank you for the remark. Control subjects were matched for age and sex. Indeed, 37.9% of patients vs. 19.7% of controls were married; however, more controls than patients were in a partner relationship, and the difference between the percentage of single individuals was not very significant (51.5% of patients vs. 63.6% of controls). We have added to the Limitations section that the participants were not matched according to their relationship status.
  3. Thank you for this comment. We have chosen to analyze the whole group, including both the nIHH and KS patients since we work with such patients daily in our clinical practice. The sense of smell may enrich sexual experience in general [1,2]; therefore, we wanted to keep this division. We had expected worse results from the psychological part of the questionnaire in the KS vs. nIHH group. However, no statistically significant differences were found between the nIHH and the KS patients based on the analyzed sociodemographic and clinical factors and the questionnaire results. We believe that excluding a group of 23 nIHH individuals would be a loss of important data and a further reduction of the study sample. There is the risk that when a sample size would be small, only large effects would be observed, leading to a potential overestimation of the true effect and, therefore, further uncertainties around any conclusions[3]. A separate analysis of KS patients is available in Table 2b, where we have added a column to include the comparison between KS subjects and controls to improve the value of this study further.
  4. Following your advice, the discussion was shortened and streamlined as much as possible.
  5. Thank you for this suggestion. The sentence has now been changed to improve its clarity. ("Compared with a large survey of US adults in which 5.4% of men reported never being sexually active, as many as 26% of IHH men reported sexual inactivity in another study (p<0.001)").

References

  1. Bendas, J.; Hummel, T.; Croy, I. Olfactory Function Relates to Sexual Experience in Adults. Arch Sex Behav 2018, 47, 1333-1339, doi:10.1007/s10508-018-1203-x.
  2. Schafer, L.; Mehler, L.; Hahner, A.; Walliczek, U.; Hummel, T.; Croy, I. Sexual desire after olfactory loss: Quantitative and qualitative reports of patients with smell disorders. Physiol Behav 2019, 201, 64-69, doi:10.1016/j.physbeh.2018.12.020.
  3. Button, K.S.; Ioannidis, J.P.; Mokrysz, C.; Nosek, B.A.; Flint, J.; Robinson, E.S.; Munafo, M.R. Power failure: why small sample size undermines the reliability of neuroscience. Nat Rev Neurosci 2013, 14, 365-376, doi:10.1038/nrn3475.

Round 2

Reviewer 2 Report

The authors referred point-to-point to my previous comments. In fact, the authors seem to agree with the basic methodological limitations regarding the study population and the choice of controls noted in my first review, which they intend to correct in a planned subsequent study.

  1. Table 2a: mobility, vision, hearing, and breathing look similar in the IHH and controls, which doesn’t seem compatible with the noted highly significant p-values. I’ve skipped this in my first review).
  2. The discussion is still cumbersome.

Author Response

Thank you very much for all your comments. We truly believe we took up in the manuscript the crucial issues that are often not fully recognized.

  1. The data in the Table 2a is correct. A Mann–Whitney test was used to compare the surveys data between patients and controls. The Mann-Whitney test is an alternative to a t-test when the data are not normally distributed (skewed). The test can detect not only the differences in medians but also differences in shape and spread. Differences in population medians are often accompanied by equally important differences in shape. It is not uncommon that the medians are equal, and the data distributions are entirely different. Differences in spread and shape are indirect reflect in differences between interquartile ranges (IQRs). An example of similar data is attached. The picture shows an example of equal medians and different data distributions.

  1. Following your advice, we changed the discussion to improve its clarity and fluency.

Once again, thank you for your detailed and accurate review.
